# Alternative Environmentally Friendly Insulating Gases for SF$_6$

**Yong Wang, Danqing Huang, Jing Liu \*, Yaru Zhang and Lian Zeng**

Electric Power Test and Research Institute, Guangzhou Power Supply Co. Ltd., Guangzhou 510410, China; wangy@guangzhou.csg.cn (Y.W.); xujie@guangzhou.csg.cn (D.H.); zhangyaru1989@163.com (Y.Z.); whu282070193@live.com (L.Z.)

**\*** Correspondence: greengasguangzhou@163.com

**Abstract:** Sulfur hexafluoride (SF$_6$) shows excellent insulation performance as an insulating gas. It is suitable for various climate conditions due to its low boiling point (−64 °C). Therefore, it has been widely used in power grid equipment. However, its global warming potential (GWP) is 23,500 times higher than that of CO$_2$. Thus, it is imperative to find an environmentally friendly insulating gas with excellent insulation performance, lower GWP, and which is harmless to equipment and workers to replace SF$_6$. In this review, four possible alternatives, including perfluorocarbons, trifluoroiodomethane, perfluorinated ketones, and fluoronitrile are reviewed in terms of basic physicochemical properties, insulation properties, decomposition properties, and compatibility with metals. The influences of trace H$_2$O or O$_2$ on their insulation performances are also discussed. The insulation strengths of these insulating gases were comparable to or higher than that of SF$_6$. The GWPs of these insulating gases were lower than that of SF$_6$. Due to their relatively high boiling point, they should be used as a mixture with buffering gases with low boiling points. Based on these four characteristics, perfluorinated ketones (C$_5$F$_{10}$O and C$_6$F$_{12}$O) and fluoronitrile (C$_4$F$_7$N) could partially substitute SF$_6$ in some electrical equipment. Finally, some future needs and perspectives of environmentally friendly insulating gases are addressed for further studies.

**Keywords:** SF$_6$; environmentally friendly insulating gas; perfluorocarbon; trifluoroiodomethane; perfluorinated ketone; fluoronitrile

## 1. Introduction

In high-voltage transmission systems, gas insulation has the advantages of being light weight, cost-effective, having simple manufacturing construction, and recyclability when compared with liquid or solid insulation. Thus, it has been widely applied in power grids all over the world. At first, a mixture of CCl$_4$ vapor and air was used as insulating gas. Herb and Rodine found that CCl$_4$ vapor and air can synergize and enhance the dielectric strength, especially with a lower CCl$_4$ concentration [1]. Charlton and Cooper found that CCl$_2$F$_2$ and CF$_4$ also showed better dielectric strength than N$_2$ [2,3]. SF$_6$ was first patented as an insulating gas by Cooper in 1938 (Figure 1). Since then, it has been studied systematically. It has been noted for its arc quenching capability and insulating properties. The excellent arc quenching capability is because of its high heat capacity, dissociation, and reassembly properties. The high dielectric strength can be attributed to its large molecular weight, complexity, and electron affinity, which affects the reaction between gas molecules and free electrons [4]. The decomposed products of SF$_6$ can recompose again when the temperature decreases, which ensures that the insulation strength is maintained well. As a result, it decomposes by only about 5% after working at 140 °C for 25 years [5]. Besides, it is non-poisonous, chemically stable, and non-flammable, which provides security for operation in practical applications. Considering the dielectric strength, cost, stability,

toxicity, and liquefaction temperature, SF$_6$ stands out as the best insulating gas. It has been widely used in air-insulated switchgear (AIS) and gas-insulated switchgear (GIS) since the 1960s [6].

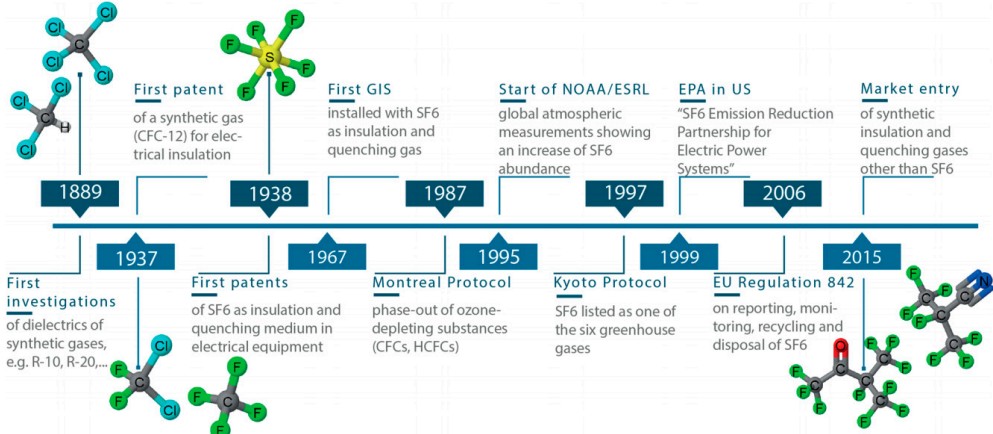

**Figure 1.** Development of insulating gases [7].

However, SF$_6$ has also caused serious environmental problems. It was identified as one of the seven greenhouse gases in the Kyoto Protocol. It shows a remarkable absorption at infrared frequency and absorbs upward radiance 42,000 times more effectively than CO$_2$, thus causing a great greenhouse effect [7]. The global warming potential (GWP) of SF$_6$ is 23,500 times higher than that of CO$_2$ over a 100 year integration time horizon according to the report of Intergovernmental Panel on Climate Change (IPCC) in 2013, and its lifetime in the atmosphere reached to 850 years with an uncertainty range of 580–1400 years [8]. With the rapid development of electrical insulation media, large amounts of SF$_6$ have been leaking or discharged into the atmosphere. In fact, the concentration of SF$_6$ in the atmosphere increased by 20% from 2010 to 2015 (Figure 2) [9]. It was estimated that the emissions will reach 4270 ± 1020 t in 2020 [10]. The greenhouse effect caused by SF$_6$ will be incalculable. Besides, when water vapor exists in the insulating equipment containing SF$_6$, the reaction generates SOF$_4$, SO$_2$F$_2$, S$_2$F$_{10}$, SF$_4$, HF, and SO$_2$. Among the products, SO$_2$F$_2$, S$_2$F$_{10}$, and SF$_4$ are highly toxic. HF and SO$_2$ are corrosive for insulating equipment [11]. Thus, the use and emission of SF$_6$ should be seriously restricted. One approach to reduce the emission of SF$_6$ was to replace some of the SF$_6$ with other inert substances with lower GWPs. For example, SF$_6$/N$_2$ was selected. Although the dosage of SF$_6$ is decreased, the GWP of a SF$_6$(10%)/N$_2$ mixture by volume is still unacceptable, at 8650 [9].

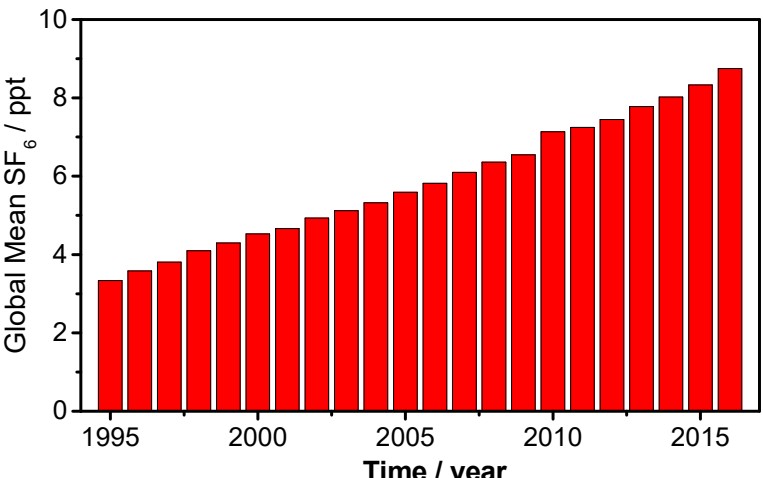

**Figure 2.** The concentration of SF$_6$ in the atmosphere since 1995.

Therefore, an environmental alternative to completely replace $SF_6$ is necessary and urgent. It should meet the features of low GWP, no ozone depletion potential (ODP), it should be non-toxic or hypotoxic, and have high dielectric strength, good thermal conductivity, low boiling point, good compatibility with switchgear materials, etc. [5,9]. Herein, we have reviewed the pioneered studies about environmentally friendly insulating gases, including perfluorocarbon, trifluoroiodomethane, perfluorinated ketones, and fluoronitrile (Table 1). For each alternative, its basic physicochemical properties, insulation properties, decomposition properties, metal compatibility, and influence of trace $H_2O$ or $O_2$ on its insulation performance are reviewed in detail in order to provide a better understanding of these compounds' insulation performances.

**Table 1.** Basic properties of compounds used in electrical insulation. GWP: global warming potential; ODP: ozone depletion potential.

| Chemical Formula | GWP/ 100-Years | Lifetime/ Years | Dielectric Strength Relative to SF$_6$ | Boiling Point/ °C | Toxicity | ODP | Flammability |
|---|---|---|---|---|---|---|---|
| $SF_6$ | 22,800 | 850 | 1 | −64 | Non-toxic | 0 | Non-flammable |
| $CF_4$ | 9200 | 50,000 | 0.4 | −128 | Low-toxicity | 0 | Non-flammable |
| $C_2F_6$ | 12,200 | 10,000 | 0.76 | −78.1 | Non-toxic | 0 | Non-flammable |
| $C_3F_8$ | 8830 | 2600 | 1.01 | −36.7 | Non-toxic | 0 | Non-flammable |
| $c\text{-}C_4F_8$ | 8700 | 3200 | 1.3 | −8 | Non-toxic | 0 | Non-flammable |
| $CF_3I$ | 0.4 | 0.0055 | 1.23 | −22 | Non-toxic | 0 | Non-flammable |
| $C_5F_{10}O$ | 1 | 0.044 | 1.5–2 | 27 | Non-toxic | 0 | Non-flammable |
| $C_6F_{12}O$ | 1 | 0.014 | 2.7 | 49 | Non-toxic | 0 | Non-flammable |
| $C_4F_7N$ | 2100 | 22 | 2 | −4.7 | Non-toxic | 0 | Non-flammable |
| $CF_4$ | 6630 | 50,000 | 0.4 | −128 | - | - | Non-flammable |
| $CO_2$ | 1 | - | 0.32–0.37 | −79 | Non-toxic | 0 | Non-flammable |
| $N_2$ | - | - | 0.34–0.43 | −196 | Non-toxic | 0 | Non-flammable |
| air | - | - | 0.37–0.40 | −193 | Non-toxic | 0 | Non-flammable |
| He | - | - | 0.02–0.06 | −268.9 | Non-toxic | 0 | Non-flammable |
| Ar | - | - | 0.04–0.10 | −186 | Non-toxic | 0 | Non-flammable |

## 2. Perfluorocarbons

Due to the electronegativity of fluorine, it is believed that perfluorocarbon has good insulation performance. Therefore, perfluorocarbons have attracted a great deal of attention as new insulation gases. The mainly proposed perfluorocarbons are $CF_4$, $C_2F_6$, $C_3F_8$, and $C_4F_8$. Their basic properties are listed in Table 1. Their GWPs are all lower than that of $SF_6$, they show no ozone depletion potential, comparable dielectric strength, and relatively lower GWPs relative to $SF_6$.

### 2.1. Perfluoromethane (CF$_4$), Perfluoroethane (C$_2$F$_6$), and Perfluoropropane (C$_3$F$_8$)

$CF_4$, $C_2F_6$, and $C_3F_8$ have the potential to be used in gas insulation equipment because of their strong electronegative property. However, the lifetimes of $CF_4$ and $C_2F_6$ are as long as 50,000 and 10,000 years, respectively. Their dielectric strengths are both lower than that of $SF_6$. Further, $CF_4$ may cause choking disease. Therefore, $CF_4$ and $C_2F_6$ are unsuitable for gas insulation.

Meanwhile, $C_3F_8$ is harmless to $O_3$ in the stratosphere. It also has low toxicity, good thermal stability, relatively low boiling point, and comparable dielectric strength to $SF_6$. The GWP of $C_3F_8$ is 8830, which is 38.7% that of $SF_6$. The breakdown voltages of $C_3F_8/N_2$ or $C_3F_8/CO_2$ have a significant linear correlation with the ratio of $C_3F_8$. The GWPs of $C_3F_8$ (12%)/$N_2$ (2736) and $C_3F_8$ (12%)/$CO_2$ (6612) were found to be 12% and 29% of that of $SF_6$ (22,800), respectively [12]. The $C_3F_8/N_2$ mixture exhibited higher dielectric strength than that of $C_3F_8/CO_2$. When the ratio of $C_3F_8$ was 20%, the insulation strength of $C_3F_8/N_2$ reached 60% of that of pure $C_3F_8$. The insulation strength under 0.79 MPa was comparable to that of $SF_6$ at 0.5 MPa. Besides, the liquefaction temperature decreased to −30 °C and the GWP also decreased greatly [13]. Thus, it is feasible to apply $C_3F_8/N_2$ or $C_3F_8/CO_2$ in practical insulation equipment.

## 2.2. Perfluorocyclobutane (c-C$_4$F$_8$)

Among CF$_4$, C$_2$F$_6$, C$_3$F$_8$, and c-C$_4$F$_8$, c-C$_4$F$_8$ exhibits the highest dielectric strength [14]. The insulation strength is about 1.3 times higher than that of SF$_6$. The GWP of c-C$_4$F$_8$ is 8700, which is 38.2% of that of SF$_6$. Moreover, c-C$_4$F$_8$ also has the features of non-toxicity, no O$_3$ destruction, and high thermal stability. Thus, it has the potential to replace SF$_6$ as an environmentally friendly insulating gas [15]. However, due to its high boiling point (−8 °C), it should be used by mixing with CF$_4$, N$_2$, CO$_2$, or air. The dielectric strength of c-C$_4$F$_8$/CO$_2$ is higher than that of SF$_6$/CO$_2$, and the GWP of c-C$_4$F$_8$/CO$_2$ is much lower than that of SF$_6$/CO$_2$ [16]. Li et al. [17] studied the dielectric strength of c-C$_4$F$_8$ with CF$_4$, CO$_2$, N$_2$, O$_2$, and air mixture by Boltzmann equation. They found that c-C$_4$F$_8$/N$_2$ and c-C$_4$F$_8$/air mixtures showed comparable dielectric strength, which were higher than those of c-C$_4$F$_8$/CF$_4$, c-C$_4$F$_8$/CO$_2$, and c-C$_4$F$_8$/O$_2$. When the concentration of c-C$_4$F$_8$ exceeded 80%, the dielectric strengths of c-C$_4$F$_8$/N$_2$ and c-C$_4$F$_8$/air were comparable to that of pure SF$_6$. After 30 experimental breakdown tests, the breakdown voltage of the c-C$_4$F$_8$/N$_2$ decreased by only 0.76%, indicating a good self-recovery characteristic. It was also reported that the decomposition rate of c-C$_4$F$_8$/N$_2$ was lower than that of pure c-C$_4$F$_8$ at the same temperature, which is more suitable in practical gas insulation systems. The main decomposition path of c-C$_4$F$_8$ was from c-C$_4$F$_8$ to C$_2$F$_4$, and it further decomposed into CF$_2$:, F·, CF$_3$·, CF·, C, CF$_4$, and C$_2$F$_4$ [18]. However, when a certain amount of O$_2$ was added into the mixture gas of c-C$_4$F$_8$/N$_2$, the breakdown voltage decreased more and more observably with the O$_2$ content increase from 0% to 1%. Then, the breakdown voltage decreased slightly when further increase of the O$_2$ content. The breakdown voltage decreased by 4.47% after 30 breakdown tests in the presence of 3% O$_2$. This was mainly attributed to the relatively lower dielectric strength of O$_2$ and the new produced products [19]. O$_2$ promotes the decomposition of c-C$_4$F$_8$ and generates the very toxic and corrosive COF$_2$. Thus c-C$_4$F$_8$ should be used without O$_2$ [20].

## 3. Trifluoroiodomethane (CF$_3$I)

CF$_3$I is a colorless, odorless, incombustible, and stable gas. Because of the excellent electronegative property of CF$_3$I, its dielectric strength is 1.2 times higher than that of SF$_6$. Besides, the GWP of CF$_3$I is 1–5, which is far less than that of SF$_6$. The C–I bond can be easily cracked under UV irradiation. Therefore, its lifetime in atmosphere is less than 2 days, and it does not cause O$_3$ destruction [21,22]. According to these characteristics, CF$_3$I has been a potential alternative to SF$_6$ as a new insulating gas.

Due to its high boiling point of −22.5 °C and the formation of I$_2$ in pure CF$_3$I, CF$_3$I should be mixed with other gases with low boiling point, such as N$_2$, CO$_2$, O$_2$, air, CF$_4$, Ar, Xe, and He. Among the mixtures, CF$_3$I/N$_2$ showed the best insulating strength [22,23]. Li et al. [22] found that the saturated vapor pressure of CF$_3$I/N$_2$ was higher than that of c-C$_4$F$_8$/N$_2$, indicating that CF$_3$I-N$_2$ can be used under higher pressure. Besides, the dielectric strength of CF$_3$I/N$_2$ was higher than that of c-C$_4$F$_8$/N$_2$, and they were both higher than that of SF$_6$/N$_2$ [24]. The dielectric strength of CF$_3$I (20%)/N$_2$ at 0.79 MPa was 102% of SF$_6$ at 0.5 MPa at −10 °C. When the CF$_3$I concentration exceeded 65%, the insulation strength of CF$_3$I/N$_2$ was higher than that of SF$_6$/N$_2$. It was even higher than that of pure SF$_6$ when CF$_3$I concentration exceeded 70% [23]. Regarding CF$_3$I/CO$_2$, the CF$_3$I and CO$_2$ can act synergistically and enhance the physicochemical properties of CF$_3$I. When the ratio of CF$_3$I or SF$_6$ was 10%–30% at 0.1–0.3 MPa, the partial discharge inception voltage of CF$_3$I/CO$_2$ was 0%–20% higher than that of SF$_6$/CO$_2$. The insulation strength of CF$_3$I/CO$_2$ was comparable or even higher than that of SF$_6$/CO$_2$ [25]. In this case, both the boiling point and insulation strength could satisfy the practical requirements. The breakdown performance of CF$_3$I/CO$_2$ was also superior to that of CF$_3$I/N$_2$. In quasi-uniform and highly non-uniform electric fields, the breakdown voltages of CF$_3$I/CO$_2$ were 84% and 65% of pure SF$_6$, which were both higher than that of CF$_3$I/N$_2$ [26,27]. The 50% breakdown voltages of CF$_3$I (30%)/CO$_2$ and CF$_3$I (20%)/CO$_2$ under 0.1 MPa were 67.1 and 66.6 kV, respectively. For CF$_3$I (30%)/N$_2$ and CF$_3$I (20%)/N$_2$, they were 60.5 and 50.1 kV, respectively [28]. After 20 breakdown experiments, less CF$_3$I decomposed in CF$_3$I/CO$_2$ mixture than that in CF$_3$I/N$_2$ mixture. It was explained that CO$_2$ could

provide an additional C source for the reaction system to maintain the C balance, which suppressed the decomposition of $CF_3I$ [26].

According to density functional theory (DFT), the reactions of $CF_3I$ to $CF_4$, $C_2F_6$, $C_2F_4$, and $C_2F_5I$ were more energetically favorable than that to $C_3F_8$, $C_3F_6$, and $I_2$. Thus, the decomposition products were mainly $C_2F_6$, $C_2F_4$, and $I_2$. It can be clearly seen that the transparent glass changed to tawny after several experiments, indicating the formation of $I_2$. The products in partial discharge were stable after 20 h test [29]. Although the products cannot reassemble to $CF_3I$ completely after discharge, there is a dynamic equilibrium among $CF_3$, $CF_2$, I, F·, and $CF_3I$. Thus, the insulating strength can be maintained well for pure $CF_3I$ [30].

However, in the presence of $O_2$, the O· from $O_2$ consumes free radicals ($CF_3$, $CF_2$:) from $CF_3I$ and generates $COF_2$ (Figure 3), which is a highly toxic irritant for respiratory mucosa and skin. What is more, it destroys the dynamic equilibrium among $CF_3$, $CF_2$, I, F·, and $CF_3I$, hindering the regeneration of $CF_3I$. As a result, the $CF_3I$ content and insulation performance decreased with the extension of discharge time [31]. In order to ensure the insulation strength and safety, the $O_2$ content in $CF_3I$ cannot exceed 7% and 20%, respectively [32]. Therefore, it is impracticable to use $O_2$ and air as buffer gases with $CF_3I$ in GIS.

**Figure 3.** Decomposition mechanism of $CH_3I$ with $O_2$ during discharge.

Moreover, the free radicals H· and HO produced from $H_2O$ destroy the balance between $CF_3I$ and free radicals, which aggravates the decomposition of $CF_3I$ and generated $C_2F_6$, $I_2$, $C_2F_4$, $C_2F_5I$, $C_3F_8$, HF, $H_2$, $COF_2$, $CF_3H$, $CF_3OH$. As a result, the partial discharge initial voltage and insulating strength decrease gradually [33,34]. So, it is vital to control the content of $H_2O$ in insulating systems.

Zhang et al. [35] studied the influence of metal particles (Cu, Al, and Fe) on the insulation property of $CF_3I$. They found the metal particles—especially Cu and Al—could increase the electrical conductivity and decrease the insulation strength of $CF_3I$. The breakdown voltages decreased with the increase of metal particles. Therefore, the metal particles in insulation equipment should be well-covered by an insulating varnish or sleeve to avoid the interaction between $CF_3I$ and metal particles.

Zhang et al. [36] studied the feasibility of $CF_3I/N_2$ in gas insulating equipment. They concluded that $CF_3I$ (30%)/$N_2$ at 0.3 MPa could be applied in some low-pressure insulating equipment. By increasing

the total pressure or the partial pressure of $CF_3I$, the $CF_3I/N_2$ can also be applied in apparatuses requiring high insulation strength. Tan et al. [37] applied a $CF_3I$ (20%)/$N_2$ mixture in 126 kV GIL (gas-insulated line). The insulation performance was 83% of that of $SF_6$ (20%)/$N_2$ and 59% of that of pure $SF_6$. When the pressure of $CF_3I$ (20%)/$N_2$ exceeded 0.7 MPa, it could meet the insulating and safety requirements. However, authors did not consider the safety and feasibility over a long time period.

## 4. Perfluorinated Ketones ($C_5F_{10}O$ and $C_6F_{12}O$)

Recently, it was found that perfluorinated ketones ($C_nF_{2n}O$:$C_5F_{10}O$ and $C_6F_{12}O$) can act as new eco-friendly and promising insulating gases. They were initially applied in fire extinguishing applications due to their incombustibility [38,39]. Their physical property parameters can be seen in Table 1. $C_nF_{2n}O$ shows high insulation capacity and its dielectric strength is 1–3 times higher than that of $SF_6$. Moreover, the atmospheric lifetime is just 7 days because of its instability under UV radiation, and it does not cause any damage to $O_3$. Therefore, it causes low greenhouse effect and other atmospheric environmental damage. However, the boiling point of $C_nF_{2n}O$ (n = 5, 6) is above 27 °C, making it easy to liquify under natural conditions. Therefore, it is infeasible to apply pure fluoroketones as insulating gases, but only as additives to other buffer gases with low boiling points, such as $N_2$, air, and $CO_2$.

### 4.1. Perfluoropentanone ($C_5F_{10}O$)

Zhang et al. [40,41] studied the decomposition mechanism of $C_5F_{10}O$ products by gas chromatography-mass spectrometry (GC-MS) and density functional theory (DFT). The reaction paths of $C_5F_{10}O$ are shown in Figure 4. According to the relative energy change, the breakage of the C–C bond between carbonyl carbon and α-carbon atom was more likely to occur and generate $CF_3CO\cdot$ and $C_3F_7$ (Reaction A1) or $C_3F_7CO$ and $CF_3$ (Reaction B1). They reacted further to generate $CF_4$, $C_2F_6$, $C_3F_8$, $C_3F_6$, $C_4F_{10}$, $C_5F_{12}$, and $C_6F_{14}$. The decomposition rate increased with the increase of breakdown tests, generating more products with weaker dielectric strength relative to $C_5F_{10}O$. The products cannot reassemble into $C_5F_{10}O$ when the environment temperature cools down [42]. As a result, the breakdown voltage decreased gradually. Besides, they also found that when the temperature was over 625 K and 825 K, the decomposition of $C_3F_7CO\cdot$ and $CF_3CO\cdot$ was enhanced. Reactions A1 and B1 would change to spontaneous. Among the products, $C_2F_6$, $C_3F_6$, and $C_4F_8$ have choking, bronchitis, anesthetic, and pneumonia effect. However, the content of $C_5F_{10}O$ in practical application is below 20% and the concentration of products is extremely low. It has been reported that during arc discharge, the product of $C_3F_6$ was 50 ppm and just 6.5 ppb may have leaked into the air, which was far less than the exposure threshold of 0.1 ppm [43]. Although the GWPs of $CF_4$, $C_2F_6$, $C_3F_8$, $C_4F_{10}$, and $C_6F_{14}$ are 7390, 12,200, 8830, 8860, and 9300 (much higher than that of $C_5F_{10}O$), it should be noted that the concentration of deposited products is extremely low under normal working conditions [44]. Therefore, $C_5F_{10}O$ is safe as an insulating gas. The application in GIS does not pose a threat to the environment or human health.

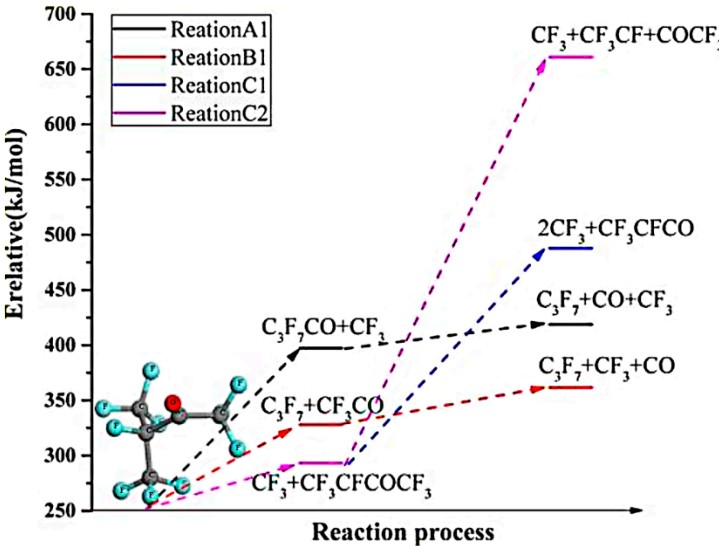

**Figure 4.** The main reaction paths of $C_5F_{10}O$ [40].

As an additive, the ratio of $C_5F_{10}O$ in mixtures is usually less than 20%. For $C_5F_{10}O/N_2$ mixtures, the GWPs are lower than 0.7. However, these mixtures generate some CO and $CF_3CN$ with high toxicity. For $C_5F_{10}O/air$ mixtures, less CO was generated and more oxygenated chemicals were generated, decreasing the toxicity of the products. However, more $C_5F_{10}O$ would decompose in $C_5F_{10}O/air$ mixtures than that in $C_5F_{10}O/N_2$ mixtures [45]. For $C_5F_{10}O/CO_2$ mixtures, the breakdown voltages of $C_5F_{10}O$ (10%)/$CO_2$ mixtures can reach to 62% of $SF_6$ under 200 kPa. When the percentage of $C_5F_{10}O$ increased to 20%, the breakdown voltage increased by 32.5% [46].

Zhang et al. [47] studied the compatibility between $C_5F_{10}O$ and Cu by theoretical calculation. Due to the high activity of the carbonyl group in $C_5F_{10}O$, it could be strongly absorbed on Cu (1 1 1) surfaces by chemical bonding. However, the interaction between the F atom and Cu is weak, which contributed to physical adsorption. Besides, they studied the compatibility between $C_5F_{10}O$ and Al or Ag. The strong interaction between $C_5F_{10}O$ and Al (1 1 1) was chemical adsorption. The weak adsorption on Ag (1 1 1) resulted from van der Waals force. Thus, they considered that Ag is more compatible with $C_5F_{10}O$ than Cu and Al [48].

### 4.2. Perfluorohexanone($C_6F_{12}O$)

When adding 3% $C_6F_{12}O$ into $N_2$, the liquefaction temperature was $-26$ °C. The breakdown voltage of the mixture gas was 1.7 times higher than that of the pure $N_2$, which was equal to that of $SF_6$ (10%)/$N_2$. The decomposed products of $C_6F_{12}O/N_2$ were mainly CO, $CO_2$, $CF_4$, $C_2F_6$, $C_2F_4$, $C_3F_8$, $C_3F_6$, $CF_3CN$, $C_2HF_5$, $C_4F_{10}$, $C_5F_{12}$, and $C_6F_{14}$. Similar to $C_5F_{10}O/N_2$, the reaction generated $CF_3CN$, which causes mortal danger [49]. Besides, the products (e.g., $C_2F_6$, $C_3F_8$, and $C_4F_{10}$) showed high insulation strength. Thus, the breakdown voltage of $C_6F_{12}O$ (3%)/$N_2$ was maintained even after 100 voltage breakdown tests. With the increase of total pressure, the breakdown voltage of the mixture gas decreased gradually [50,51]. For the mixture of $C_6F_{12}O$ and air, the generation of $CF_3CN$ is avoided. The products are mainly $CO_2$, $CF_4$, $C_2F_6$, $C_3F_8$, and $C_2O_3F_6$, among which the content of $CO_2$ is the highest [49]. When the temperature exceeded 475 °C, the decomposition of $C_6F_{12}O/CO_2$ was enhanced. The possible decomposition paths are shown in Figure 5. The strength of the C–C bond is weaker than that of C–F and C=O bonds. Thus, $C_6F_{12}O$ firstly decomposed into $C_3F_7COCF_2$ and $C_2F_5COCF_2CF_3$. Then, they further decomposed into fragments such as F, $CF_3·$, $CF_2$, CF·, $C_3F_7·$, CO, $COF_2$, and C. These free radicals cannot recombine to $C_6F_{12}O$. The final products were mainly $CF_4$, $C_2F_6$, $C_3F_8$, $C_3F_6$, and $C_5F_{12}$ and their contents decreased as follows: $C_2F_6 > C_3F_6 > C_3F_8 > CF_4$ [52].

(a) $C_6F_{12}O$

(b) $C_3F_7COCF_2$

(c) $CF_3CFC(O)(CF_2)$

**Figure 5.** The proposed decomposition mechanisms of $C_6F_{12}O/CO_2$ [52].

In the presence of trace water, the produced HO and H aggravate the decomposition of $C_5F_{10}O$ and produce more new products, such as $C_3F_7COH$, $C_3F_7OH$, HF, and $CF_2O$. The ionization parameters of

the new formed products are lower than that of $C_5F_{10}O$, thus resulting in decreased dielectric strength. Furthermore, the newly formed $CF_2O$ has an irritative effect on the skin and respiratory mucosa. HF can cause aggressive corrosion to equipment and irritation to humans [53]. Therefore, the presence of water negatively impacts the insulation performance of $C_5F_{10}O$.

During practical application, the insulating gas and equipment must have good compatibility to maintain security. Zhang et al. [54] systematically studied the compatibility between $C_6F_{12}O$ and metal materials by combining experimental tests and theoretical calculation. $C_6F_{12}O$ can be absorbed on the surface of Cu and Al due to chemical adsorption between C=O and $C_6F_{12}O$, generating metal oxide. The interaction between $C_6F_{12}O$ and Ag is attributed to physical adsorption, and thus less $C_6F_{12}O$ was absorbed. Overall, from their SEM images, it was seen that $C_6F_{12}O$ did not cause serious corrosion to the surface of Cu, Al, or Ag even after reaction for 125 days. Thus, the compatibilities between $C_6F_{12}O$ and Cu, Al, and Ag were excellent.

## 5. Fluoronitrile ($C_4F_7N$)

Heptafluoro-iso-butyronitrile ($C_4F_7N$), one kind of fluoronitrile, was firstly prepared and commercialized by the 3M$^{TM}$ Company. It has the features of low toxicity and high thermal conduction. Its lifetime in the atmosphere is 22 years. The GWP is 2100, and its insulation strength is about one-fold higher than that of $SF_6$ under normal pressure. Its insulation property makes it a promising alternative to $SF_6$ in electrical insulation systems.

However, $C_4F_7N$ cannot be applied alone due to its relatively high boiling point of $-4.7$ °C. Thus, it is necessary to add other buffer gases to decrease the boiling point of insulating gas mixtures in practical applications. The breakdown voltage of $C_4F_7N$ (12%)/$N_2$ at 0.4 MPa was comparable to pure $SF_6$ at 0.2 MPa. Increasing the ratio of $C_4F_7N$ in gas mixtures can effectively enhance the insulation strength Considering the minimum temperature of $-25$ °C in practical application, the breakdown voltages of $C_4F_7N$ (5%)/$N_2$ at 0.3, 0.4, 0.5, and 0.6 MPa were 63.4%, 54.6%, 49%, and 56.4% of that of pure $SF_6$, respectively. Moreover, the negative partial discharge inception voltages reached 80.4%, 66.9%, 62.8%, and 68.8% of that of pure $SF_6$, respectively. The insulation strength of $C_4F_7N/N_2$ in a uniform electric field was higher than that in a non-uniform electric field [55]. The insulation strength and breakdown voltage of $C_4F_7N$ (5%)/$N_2$ were 83.34% of that of pure $SF_6$. The breakdown voltage was maintained at 33.6 kV after 30 breakdown tests, indicating an excellent self-recovery property. The GWP of $C_4F_7N$ (5%)/$N_2$ was less than 600, which was far less than that of $SF_6$ (22,800). The probable decomposition pathways are shown in Figure 6. $C_4F_7N$ mainly decomposed to four free radicals ($CF_3$, CN, F, and $C_3F_7$) and the path from $C_4F_7N$ to $C_3F_4N$ and $CF_3$ was the most energy favorable. The free radicals react with each other form different products. $CF_3$ can react with CN, F·, and other free radicals to generate products. Among the products, $C_2F_6$, $CF_4$, and $CF_3CN$ are dominant. Although the products (e.g., $CF_3CN$ and $C_2F_5CN$) were toxic, their concentrations were extremely low. Overall, the toxicity of the products was lower than that of the products of $SF_6$ decomposition, and was acceptable. Besides, $N_2$ was more likely to decompose than $C_4F_7N$. Thus, it acted as buffer gas and avoided the excessive decomposition of $C_4F_7N$, which ensured the insulation performance [56]. For $C_4F_7N$, the decomposed products were mainly $C_3F_7$, CN, CNF, $CF_3$, $CF_2$, CF, $CF_3CFCN(C_3NF_4)$, F, other free radicals, and $CF_4$. Their amounts increased with the increase of temperature. The free radicals recombined with each other and generated products such as $CF_4$, $C_2F_6$, $C_3F_8$, $CF_3CN$, CO, and so on. However, less $C_4F_7N$ decomposed and less products were generated in $C_4F_7N/CO_2$ mixture due to the buffer action of $CO_2$. At 2400 K, the amount of products in pure $C_4F_7N$ was 96%, while that in $C_4F_7N/CO_2$ was 58%. Besides, the amount of $CF_4$ and C decreased after introducing $CO_2$, thus avoiding the formation of precipitate carbon and other products with relatively inferior insulation performance [57,58].

**Figure 6.** Probable decomposition pathways of $C_4F_7N/N_2$ [56].

The H· and HO· radicals generated from $H_2O$ decomposition are active in reacting with free radicals decomposed from $C_4F_7N$ during discharge (Figure 7). the activation energy in all possible paths with $H_2O$ is lower than that without $H_2O$. Thus, the decomposition of $C_4F_7N$ was accelerated, and the insulation performance was weakened. With the catalysis of H, the reactions generating HF, HCN, $CF_3H$, and other small molecules were more likely to occur. In the presence of HO·, the reaction generating $CF_3OH$ occurred easily. The products, including $CF_2O$, HF, HCN $CF_3CH_2CN$, $CF_2HCN$, and $CH_2FCN$, are toxic, which would cause damage to operation personnel. HF and HCN would also cause severe corrosion to the equipment [59].

**Figure 7.** Decomposition pathways of $C_4F_7N$ in the presence of $H_2O$.

Zhang et al. [60] studied the compatibility of $C_4F_7N$ with Cu (1 1 1) and Al (1 1 1). They found that the interaction between $C_4F_7N$ and Cu (1 1 1) or Al (1 1 1) was weak. The N atom was more likely than the F atom to react with Cu (1 1 1) or Al (1 1 1) and form a weak chemical bond. So, the compatibility of $C_4F_7N$ with Cu (1 1 1) or Al (1 1 1) was good. They also studied the compatibility of decomposition products of $C_4F_7N$ with Cu (1 1 1) and Ag (1 1 1). $C_2F_5CN$, $CF_3CN$, $COF_2$, and $CF_4$ could be adsorbed on Cu (1 1 1) and Ag (1 1 1) surfaces by van der Waals force. The adsorption energies of products

on Ag (1 1 1) surface were weaker than those on Cu (1 1 1) surface. Overall, the compatibility of decomposition products of $C_4F_7N$ with Cu and Ag were excellent [61].

## 6. Challenges and Perspectives

Most of the related studies were conducted based on theoretical calculation (e.g., DFT), and few experimental studies have been done to investigate the insulation performance in practical application—especially with the existence of trace $H_2O$ or $O_2$, which is more close to reality. A certain amount of $H_2O$ or $O_2$ may lead to the severe deterioration of insulation performance.

The compatibilities between the insulating gases or their decomposed products and metals were simulated and studied based only on a certain crystal face of metals. In electrical equipment, every crystal face can be exposed to the insulating gas. Thus, the actual compatibility between the insulating gases or their decomposed products and the insulation equipment may be very complex. It is significant to consider the compatibility through systematic experiments.

Common adsorbents such as $\gamma$-$Al_2O_3$ can not only absorb the hazardous products, but also the insulating gas. It would be interesting to design a novel adsorbent which could absorb the harmful products exclusively. This could ensure the safety of employees and equipment and maintain the insulation performance.

## 7. Conclusions

The GWPs of $C_2F_6$, $C_3F_8$, and c-$C_4F_8$ are still too high to show significant advantages compared with $SF_6$. $CF_3I$ shows distinguished low GWPs and dielectric strength, however, it has been identified as a cancerogenic substance, and its stability and compatibility with the materials of electric equipment should be further studied. The GWPs of perfluorinated ketones ($C_5F_{10}O$ and $C_6F_{12}O$) and fluoronitrile ($C_4F_7N$) are low, and they show high dielectric strengths and low toxicity, and therefore they have the potential to partially replace $SF_6$ in some electric insulation equipment. However, the compatibility of these insulating gases with the equipment materials, and the leaking rate obtained by using the conventional sealing materials should also be well studied. The adsorbents used to eliminate $H_2O$ and $O_2$, which can accelerate the decomposition of the insulating gas, should also be screened or developed to ensure the safe operation of the equipment.

**Author Contributions:** Conceptualization, J.L.; investigation, L.Z. and Y.Z.; writing-original draft preparation, Y.W.; writing-review and editing, J.L.; project administration, D.H.

**Funding:** This research was funded by the project 'Study on Physical, Chemical and Insulation Properties, and Engineering Demostration of Environmental Insulating gas (I)-Project 3-Applied Feasibility Study of New Insulating Gas in Guangzhou Power Grid', numbered as GZJKJXM20170330.

**Conflicts of Interest:** The authors declare no conflict of interest.

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
