# Peer review of "Alternative Environmentally Friendly Insulating Gases for SF6"

_processes, doi:10.3390/pr7040216_

Round 1

Reviewer 1 Report

The paper presents investigations research progress of environmental friendly insulating gases. SF6 gas, used as insulating gas, presents perfect insulation performance and it is suitable for different climate conditions regarding to low boiling point. This is a reason why it is applied in power systems, especially in GIS and GIL systems as insulation medium. There is disadvantage of the gas - global warming potential, which is higher than in case of CO2 – 42000 times. That is why it is imperative to use environmental friendly insulating gas with good insulation properties, and lower. Authors present four possible alternatives from the aspects of various properties: insulation properties, decomposition properties, compatibility with metal. The breakdown voltage of gases were comparative or higher than pure SF6. Authors concluded that more studies should be conducted before application.

So, the topic is interesting. SF6 has really prefect insulation properties, much better than air, vacuum or other gas types. Anyway, crucial disadvantage of the gas is its GWP, much bigger than in case of CO2. The solution, used for many years, is used mixture of SF6 and air with different ratio. I think, the paper is interesting, if we find some value conclusions.

47 – “SF6, however, have also caused …”. I think, it should be “SF6, however, has also caused … ”.

65 – I think, Fig.1 title should be on the same page what all Figure 1.

309 – “other insulati”. Please correct.

The paper is interesting, because presents detailed analysis of eventually gases, which can be use as insulating medium instead SF6. Anyway, I have some objections to the paper: 

1. Authors, as employers of Electric Power Institute, should know, that all mentioned gases are not used now as insulating medium in electric power system, and they are not popular gases in power systems. Only mixture SF6 with air, N2, is used as alternative to pure SF6, because of disadvantage properties of pure SF6. The assessment of the paper is difficult for me. Authors made detailed analysis of some gases, but they are not used widely in power systems. More, papers mostly Mr. Zhang, X., with his theoretical analysis are cited.

2. There is no information regarding mixture SF6 and N2, widely used in power systems as insulating medium in such apparatus as GIS, GIL, switchgears, high voltage cables, and others.

General conclusions:

1. Please add some information according mentioned mixtures.

2. Please cite others authors investigations.

Author Response

The paper presents investigations research progress of environmental friendly insulating gases. SF6 gas, used as insulating gas, presents perfect insulation performance and it is suitable for different climate conditions regarding to low boiling point. This is a reason why it is applied in power systems, especially in GIS and GIL systems as insulation medium. There is disadvantage of the gas - global warming potential, which is higher than in case of CO2 – 42000 times. That is why it is imperative to use environmental friendly insulating gas with good insulation properties, and lower. Authors present four possible alternatives from the aspects of various properties: insulation properties, decomposition properties, compatibility with metal. The breakdown voltage of gases were comparative or higher than pure SF6. Authors concluded that more studies should be conducted before application.

So, the topic is interesting. SF6 has really prefect insulation properties, much better than air, vacuum or other gas types. Anyway, crucial disadvantage of the gas is its GWP, much bigger than in case of CO2. The solution, used for many years, is used mixture of SF6 and air with different ratio. I think, the paper is interesting, if we find some value conclusions.

47 – “SF6, however, have also caused …”. I think, it should be “SF6, however, has also caused … ”.

 Response: We appreciate your comments, and we revised the manuscript according to your comment.

65 – I think, Fig.1 title should be on the same page what all Figure 1.

 Response: We appreciate your comments, and we revised the manuscript according to your comment.

309 – “other insulati”. Please correct.

 Response: We appreciate your comments, and we revised the manuscript according to your comment.

The paper is interesting, because presents detailed analysis of eventually gases, which can be use as insulating medium instead SF6. Anyway, I have some objections to the paper: 

1. Authors, as employers of Electric Power Institute, should know, that all mentioned gases are not used now as insulating medium in electric power system, and they are not popular gases in power systems. Only mixture SF6 with air, N2, is used as alternative to pure SF6, because of disadvantage properties of pure SF6. The assessment of the paper is difficult for me. Authors made detailed analysis of some gases, but they are not used widely in power systems. More, papers mostly Mr. Zhang, X., with his theoretical analysis are cited.

2. There is no information regarding mixture SF6 and N2, widely used in power systems as insulating medium in such apparatus as GIS, GIL, switchgears, high voltage cables, and others.

 General conclusions:

1.     Please add some information according mentioned mixtures.

Response: We appreciate your comment. In the text Line 64-68, One approach to reduce the emission of SF6 was to replace partial SF6 with other inert substances with lower GWPs. For example, SF6/N2 were selected. Although the dosage of SF6 decreased, the GWP of a SF6(10%)/N2 mixture by volume reached 8650 [9]. It was still unacceptable.

 We also pointed out the utilization of SF6/N2 mixture, however, since it is not the key point of this review, therefore, we did not extend the discussion on this point.

2. Please cite others authors investigations.

Response: Prof. Zhang leads a large group focusing on studying the new insulating gases, therefore, a lot of papers from that group were published in recently years, especially on theoretical study. We searched the literature databases, and we cited most of the papers related, not only citing Prof. Zhang’s work selectively.

Reviewer 2 Report

1.) The abstract should include the main findings of this paper, very briefly. Some of the sentences in the abstract is redundant. Length reduction is advised.

2.) Conclusions should be provided in Section 4, not in Section 3. Challenges and Perspectives should be included in a discussion section before the final conclusions.

3.) The conclusion must be revised in order to specify the new added scientific value of this review paper. What are the future research items suggested by this review paper?

4.) Section 2, subsections 2.1, 2.2, 2.3, 2.4 and 2.5 include a lot of information (text). They should be separated into smaller paragraphs. A dedicated overview is at advised in the beginning of Section 2. Moreover, the information must be put in a way to attract more readership. More tables and comparative analysis are advised. 

5.) The average reader should get a quick overview of how the insulating gases are used in todays applications (components and systems of "practical insulation equipment"). Clarify high voltage applications.

6.) Compare partially replacing SF6 and fully replacing SF6. Have a look at the alternatives if SF6 is to be fully replaced. Provide more data from recent publications.

Please address the following issues. 

Author Response

1.)    The abstract should include the main findings of this paper, very briefly. Some of the sentences in the abstract is redundant. Length reduction is advised.

Response: We appreciate the review’s comments, and we revised the abstract carefully according to this comments, the revised sentences were high lighted.

2.)    Conclusions should be provided in Section 4, not in Section 3. Challenges and Perspectives should be included in a discussion section before the final conclusions.

Response: We appreciate this comments. We revised the manuscript according to this comments.

3.)    The conclusion must be revised in order to specify the new added scientific value of this review paper. What are the future research items suggested by this review paper?

Response: We rewrote the conclusion section, and new added scientific value of this review paper were supplemented. The future research items suggested by this review paper could be studying the compatibility of these insulating gases with the equipment materials, and the leaking rate by using the conventional sealing materials. The adsorbants used to eliminate moisture and oxygen in the gases should also be screened and developped properly.

4.)    Section 2, subsections 2.1, 2.2, 2.3, 2.4 and 2.5 include a lot of information (text). They should be separated into smaller paragraphs. A dedicated overview is at advised in the beginning of Section 2. Moreover, the information must be put in a way to attract more readership. More tables and comparative analysis are advised. 

Response: Thank you for your comment. We organized this review paper according to the insulating gases, and mainly including the aspects of basic physicochemical properties, insulation properties, decomposition properties, compatibility with metals. Therefore, we believe it is easier to get information for the readers based on the insulating gases. Besides, in order to get an overview of the reviewed insulating gases, in table 1, we listed the basic properties of compounds used in electrical insulation.

5.)    The average reader should get a quick overview of how the insulating gases are used in todays applications (components and systems of "practical insulation equipment"). Clarify high voltage applications.

Response: Thank you for your comment. These new insulating gases were mainly on the stage of feasibility study, and not widely used in high voltage applications, therefore, we are sorry that we are unable to offer these application data.

6.)    Compare partially replacing SF6 and fully replacing SF6. Have a look at the alternatives if SF6 is to be fully replaced. Provide more data from recent publications.

Response: Thank you for your comments. To be partially replacing SF6 or fully replacing SF6 is determined by many factors. Based on their characteristics, we think partially replace SF6 is pronounced, and we also and this work is focusing on the research progress of new insulating gases.

Round 2

Reviewer 1 Report

The paper presents investigations research progress of environmental friendly insulating gases. SF6 gas, used as insulating gas, presents perfect insulation performance and it is suitable for different climate conditions regarding to low boiling point. This is a reason why it is applied in power systems, especially in GIS and GIL systems as insulation medium. There is disadvantage of the gas - global warming potential, which is higher than in case of CO2 – 42000 times. That is why it is imperative to use environmental friendly insulating gas with good insulation properties, and lower. Authors present four possible alternatives from the aspects of various properties: insulation properties, decomposition properties, compatibility with metal. The breakdown voltage of gases were comparative or higher than pure SF6. Authors concluded that more studies should be conducted before application.

Authors included all my grama sugestions in the paper text. from that point of view, the paper is ready to be published in present form.

Authors answer my scientific two questions and comments. They extended some parts of the paper including extra information about mixtures SF6 and N2.

Generally, I think, the paper is ready to be published in present form.

Author Response

Thank you for your comments!

Reviewer 2 Report

The reviewer thank the authors for the improvements.

However, they did not fully clear all of the concerns:

- The authors did not address the developments or alternatives for full replacement of SF6.

- The gases that are under feasibility study and the one that are implemented in real-world applications must be clearly distinguished in the paper (in the table and in the text).

- No improvement has been made for sections 2.1 to 2.4. This needed to improve the readability of the paper

Author Response

Dear Reviewer,

   We appreciate your time on reviewing our work. We have revised our manuscript carefully according to your comments, and the responses are listed below point by point.

Best Regards!

    Honghui Yang

#Comments and Responses:

However, they did not fully clear all of the concerns:

- The authors did not address the developments or alternatives for full replacement of SF6.

- The gases that are under feasibility study and the one that are implemented in real-world applications must be clearly distinguished in the paper (in the table and in the text).

- No improvement has been made for sections 2.1 to 2.4. This needed to improve the readability of the paper

1.)The authors did not address the developments or alternatives for full replacement of SF6.

Response: Thank you for your comments. In the introduction section, the development of insulation gas and the alternatives has been described. Insulation gas, including perfluorocarbon, trifluoroiodomethane, perfluorinated ketones and fluoronitrile, are reviewed to replace SF6. However, currently, It is not ready to full replace SF6.

2.)The gases that are under feasibility study and the one that are implemented in real-world applications must be clearly distinguished in the paper (in the table and in the text).

Response: Thank you for your comments. As far as we know, most insulation gas in practical application is still SF6. But it caused serious greenhouse effect. So much effort has been paid to study environmental-friendly insulation gas to replace SF6. The related studies are mainly in laboratory stage. Few insulation gases have been applied practically. Therefore, most studies reviewed in this manuscript are fundamental research. Some studies in real-world applications are in discussed a separate paragraph. We listed a column in table to point out which is used in real-word, and which are under feasibility study.

3.)No improvement has been made for sections 2.1 to 2.4. This needed to improve the readability of the paper.

Response: We appreciate your comment. Sections 2.1 to 2.4 have been separated into smaller paragraphs according to their classifications. The manuscript has been reorganized to improve the readability.